# Regulation of Epigenetic Modifications in the Placenta during Preeclampsia: PPARγ Influences H3K4me3 and H3K9ac in Extravillous Trophoblast Cells

**DOI:** 10.3390/ijms222212469

**Published:** 2021-11-18

**Authors:** Sarah Meister, Laura Hahn, Susanne Beyer, Corinna Paul, Sophie Mitter, Christina Kuhn, Viktoria von Schönfeldt, Stefanie Corradini, Kritika Sudan, Christian Schulz, Theresa Maria Kolben, Sven Mahner, Udo Jeschke, Thomas Kolben

**Affiliations:** 1Department of Gynecology and Obstetrics, University Hospital, LMU Munich, Marchioninistr. 15, 81377 Munich, Germany; laura.hahn@med.uni-muenchen.de (L.H.); susanne.beyer@med.uni-muenchen.de (S.B.); corinna.paul@med.uni-muenchen.de (C.P.); sophie.mitter@med.uni-muenchen.de (S.M.); viktoria.schoenfeldt@med.uni-muenchen.de (V.v.S.); theresa.kolben@med.uni-muenchen.de (T.M.K.); sven.mahner@med.uni-muenchen.de (S.M.); thomas.kolben@med.uni-muenchen.de (T.K.); 2Department of Gynecology and Obstetrics, University Hospital Augsburg, 86156 Augsburg, Germany; Christina.kuhn@uk-augsburg.de; 3Department of Radiation Oncology, University Hospital, LMU Munich, Marchioninistr. 15, 81377 Munich, Germany; stefanie.corradini@med.uni-muenchen.de; 4Medizinische Klinik und Poliklinik I, Klinikum der Universität München LMU Munich, Marchioninistr. 15, 81377 Munich, Germany; kritika.sudan@med.uni-muenchen.de (K.S.); christian.schulz@med.uni-muenchen.de (C.S.)

**Keywords:** preeclampsia, histone modification, H3K4me3, H3K9ac, PPARγ, RxRα, placenta

## Abstract

The aim of this study was to analyze the expression of peroxisome proliferator-activated receptor γ (PPARγ) and retinoid X receptor α (RxRα), a binding heterodimer playing a pivotal role in the successful trophoblast invasion, in the placental tissue of preeclamptic patients. Furthermore, we aimed to characterize a possible interaction between PPARγ and H3K4me3 (trimethylated lysine 4 of the histone H3), respectively H3K9ac (acetylated lysine 9 of the histone H3), to illuminate the role of histone modifications in a defective trophoblast invasion in preeclampsia (PE). Therefore, the expression of PPARγ and RxRα was analyzed in 26 PE and 25 control placentas by immunohistochemical peroxidase staining, as well as the co-expression with H3K4me3 and H3K9ac by double immunofluorescence staining. Further, the effect of a specific PPARγ-agonist (Ciglitazone) and PPARγ-antagonist (T0070907) on the histone modifications H3K9ac and H3K4me3 was analyzed in vitro. In PE placentas, we found a reduced expression of PPARγ and RxRα and a reduced co-expression with H3K4me3 and H3K9ac in the extravillous trophoblast (EVT). Furthermore, with the PPARγ-antagonist treated human villous trophoblast (HVT) cells and primary isolated EVT cells showed higher levels of the histone modification proteins whereas treatment with the PPARγ-agonist reduced respective histone modifications. Our results show that the stimulation of PPARγ-activity leads to a reduction of H3K4me3 and H3K9ac in trophoblast cells, but paradoxically decreases the nuclear PPARγ expression. As the importance of PPARγ, being involved in a successful trophoblast invasion has already been investigated, our results reveal a pathophysiologic connection between PPARγ and the epigenetic modulation via H3K4me3 and H3K9ac in PE.

## 1. Introduction

Affecting around five percent of all pregnancies, preeclampsia (PE) represents one of the most frequent gestational diseases [1] and a relevant cause for maternal deaths [2,3,4]. PE is defined as new-onset hypertension in pregnancy (>140/90 mm Hg) combined with organ dysfunction after the 20th week of gestation, most commonly in the sense of pathologic proteinuria (>300 mg/L) [5].

The pathophysiology of PE is currently not completely understood. Additionally, inflammatory processes [1,6,7,8] and loss of maternal tolerance to the fetus [9,10,11], maternal cardiovascular maladaptation [12,13] and placental insufficiency [14] appear to be involved in the development of PE. Impaired trophoblast invasion [15], impaired remodeling of spiral arteries [5,16], and defective placentation [13,17,18,19] contribute to the aforementioned placental insufficiency.

The process of healthy placentation is complex and contains not only vascular remodeling but a complicated process of cell differentiation and cell growth. Considering placental maturing, it is necessary to distinguish between the development of the placental villi (cytotrophoblast proliferation, syncytial fusion) [20] and the trophoblast invasion happening simultaneously to the vascular remodeling [21,22]. The trophoblast invasion begins before the 6th week of pregnancy and happens at the basal plate and the placental bed. All trophoblasts, which reside outside the placental villi are summarized under the term EVT [21]. These trophoblasts migrate together with uteroplacental arteries into the decidua, and it is already known that several instances, such as trophoblast apoptosis, lead to a defective trophoblast invasion in PE [23,24,25].

Peroxisome proliferator-activated receptors (PPARs), are transcription factors and members of the ligand-activated nuclear hormone receptor superfamily, being involved in energy metabolism, inflammation, and cell development [26,27,28,29]. After ligand binding, PPARs form heterodimers with retinoid X receptors (RXRs) and bind to PPAR-response elements (PPREs) of their target genes to activate transcription [26].

PPARγ and RxRα are predominantly known for their important role in adipogenesis and metabolism [27,28]. They are furthermore involved in trophoblast differentiation [30], placentation [31], and differentiation of the syncytium [32]. It has been shown that PPARγ stimulates trophoblast proliferation [33] and plays an important role in trophoblast invasion [34,35]. A lack of PPARγ leads to placental defects [36,37], an increase of proinflammatory cytokines [38] is associated with hypertension [36,39]. Placental PPARγ is produced by trophoblasts and endometrial stromal cells [30,40,41] and is known to be reduced in preeclampsia [32,42].

Even though PPARs are nuclear receptors, there is some evidence for an additional cytoplasmic expression [42,43,44,45]. The role of this cytoplasmic expression has not yet been definitively determined, which is why we focused on nuclear PPARγ expression.

Various epigenetic changes have been detected in PE-affected pregnancies such as DNA methylation, non-coding RNAs, and genomic imprinting [46]. Histone modification is a further epigenetic alteration. Histones can be modified by acetylation, methylation, or phosphorylation [47] adjusting the accessibility of the DNA which is wrapped around the histones [14].

Histone modification has been shown to regulate factors that are important for trophoblast invasion and migration which is defective in PE [1,46,48]. Recent data of our group showed a decrease of trimethylated lysine 4 of the histone H3 (H3K4me3) and acetylated lysine 9 of the histone H3 (H3K9ac) in EVT cells of placentas of preeclamptic mothers [49]. As H3K4me3 and H3K9ac are known to affect syncytialisation as a necessary precondition for sufficient placentation [50] there might be involvement of histone modification in trophoblast invasion.

Furthermore, there are associations between PPARγ and histone modifications. In the adipogenesis and late adipocyte differentiation, a positive influence of H3K4me3 and H3K9ac on PPARy has already been shown [51,52,53].

Since pathophysiological mechanisms of preeclampsia are not fully elucidated and the disease is hard to investigate, there is no cause-specific therapy except for delivering the baby and the placenta [4,31,54], leading to a higher rate of preterm-births and further morbidities [55]. Several molecular mechanisms still need to be investigated in the pathophysiology of PE, especially controlling gene expression, to hopefully discover a potential therapeutic target. Therefore, a further investigation of PPARy and the connection with the histone modifications H3K4me3 and H3K9ac could be illuminating regarding therapy possibilities of PE.

As ineffective trophoblast invasion represents a pivotal element of the pathophysiology of PE, the aim of this study was to analyze the expression of PPARγ and RxRα, especially in the EVT in the placenta of preeclamptic patients, and to characterize a possible association between the histone modifications found to be decreased in the placenta during PE.

## 2. Results

### 2.1. Expression of PPARγ and RxRα Is Decreased in the Decidua of PE Placentas

PPARγ expression was significantly lower in the decidua of PE placentas 4.56 ± 2.725) compared to controls (7.43 ± 3.727) (*p* = 0.004) (Figure 1B–D). Mean IRS in the syncytiotrophoblast of PE placentas (4.00 ± 2.683) and controls (4.74 ± 3.441) was not significantly different (Figure 1A,C,D).

As PPARγ builds heterodimer complexes with RxRα, the expression of RxRα was analyzed as well. The mean IRS of RxRα in the syncytiotrophoblast and the decidua was significantly diminished in PE placentas (IRS_syn_ = 4.43 ± 2.694; IRS_dec_ = 4.40 ± 2.703) compared to the controls (IRS_syn_ = 6.00 ± 2.322; IRS_dec_ = 7.22 ± 4.011) (*p*_syn_ = 0.045, *p*_dec_ = 0.038) (Figure 2A–D).

### 2.2. Correlation of H3K4me3 and H3K9ac with PPARγ and RxRα

A correlation analysis of the IRS of H3K4me3/H3K9ac—published earlier by our group [49] and the expression of PPARγ and RxRα was performed, to outline a possible connection between the transcription modulating PPARγ and RxRα and the histone modifications H3K4me3 and H3K9ac.

Both the acetylated histone H3K9ac as well as the trimethylated histone H3K4me3 correlated significantly positive with RxRα. These significant correlations were found in the syncytium (H3K9ac: r = 0.428, *p* = 0.003; H3K4me3: r = 0.448, *p* = 0.002) as well as in the decidua (H3K9ac: r = 0.464, *p* = 0.002; H3K4me3: r = 0.327, *p* = 0.032).

Furthermore, PPARγ correlated positively with the investigated histone modifications. Significant correlation of H3K9ac and PPARγ in the syncytiotrophoblast (r = 0.284, *p* = 0.046) as well as in the decidua (r = 0.399, *p* = 0.004) were found. Moreover, H3K4me3 in the syncytium correlated significantly positive with PPARγ in the decidua (r = 0.357, *p* = 0.012).

The overall results of the statistical correlation analysis of the histone modifications and PPARγ respectively RxRα are shown in Table 1.

### 2.3. Identification of Decidual Cells Expressing PPARγ/RxRα

To clarify the type of decidual cells expressing PPARγ and RxRα an immunofluorescence double staining with CK 7 was performed. Trophoblasts—in the case of the decidua—EVTs, are expressing CK 7. Therefore, decidual cells stained by CK 7 can be classified as EVTs whereas CK 7 negative tissue cells are considered to be decidual stroma cells.

The immunofluorescence double staining of PPARγ and CK 7 showed a clear double expression in EVTs, in control samples and PE samples (Figure 3A,B).

In a second step, double staining of PPARγ and RxRα was performed. The purpose of this staining was to show the retention of the known heterodimerization of PPARγ and RxRα in preeclamptic placentas (Figure 3C,D).

### 2.4. Co-Expression of H3K4me3 and H3K9ac with PPARγ and RxRα

Double immunofluorescence staining was performed to verify whether the histone modifications H3K4me3 and H3K9ac are located in the same cell type as PPARγ, respectively, RxRα. H3K4me3 respectively H3K9ac are presented in red and PPARγ, respectively, RxRα in green. Colocalization is shown as yellow color in the triple filter excitation.

Co-expression appeared in all three analyzed cell types-stromal cells, EVTs and the syncytiotrophoblast, both in control and PE placentas. A distinctly and visibly reduced amount of double-positive cells for H3K9ac respectively H3K4me3 and PPARγ was found in PE (H3K9ac + PPARγ: Control 80–90%, PE 70%, (Figure 4) H3K4me3 + PPARγ: Control 90%, PE 70% (Figure 5)); The difference between control and PE was similar for decidual cells, EVTs, and the syncytium.

Further, colocalization of H3K4me3 respectively H3K9ac and RxRα was analyzed. We found a decreased amount of double-stained cells in PE placentas compared to control placentas (Appendix A).

Concerning H3K4me3 (H3K4me + RxRα decidua: Control 40%, PE 10%; H3K4me3 + RxRα syncytiotrophoblast: Control 60%, PE 10%) the effect seemed to be slightly more pronounced in the decidua and syncytiotrophoblast compared to H3K9ac (H3K9ac + RxRα decidua: Control 50%, PE 30%; H3K9ac + RxRα syncytiotrophoblast: Control 50%, PE 30%; H3K9ac + RxRα EVT: Control 80%, PE 60%).

### 2.5. Specific PPARγ Agonist and Antagonist Regulate H3K4me3 and H3K9ac in Human Trophoblast Cells

To investigate the individual effect of PPARγ on histone modification H3K4me3 and H3K9ac in trophoblasts, the PPARγ agonist Ciglitazone and the PPARγ antagonist T0070907 were used. After 24 h incubation of HVT cells and primary isolated EVT cells with Ciglitazone and T0070907, the expression of histone modifications and PPARγ were evaluated by double immunofluorescence staining. PPARγ expression was reduced in HVT cells after incubation with the PPARγ agonist Ciglitazone and induced by the PPARγ antagonist T0070907 (Figure 5, Appendix A). The mean fluorescence intensity of H3K4me3 was significantly decreased after incubation with Ciglitazone (*p* = 0.04) and increased after incubation with T0070907 (*p* = 0.02; Figure 6A,B). The same effect could be detected in the mean fluorescence intensity of H3K9ac (*p*_Ciglitazone_ = 0.005, *p*_T0070907_ = 0.004; Figure 6C,D).

For the primary isolated EVT cells, a significant decrease of the mean fluorescence intensity of H3K4me3 and H3K9ac after adding the PPARγ agonist Ciglitazone (*p*_H3K4me3_ = 0.0143, *p*_H3K9ac_ = 0.004) could be detected. On the other hand, the PPARγ antagonist T0070907 increased H3K4me3 respectively H3K9ac (*p*_H3K4me3_ = 0.0065, *p*_H3K9ac_ = 0.007) in EVT cells (Figure 7A–D).

## 3. Discussion

Several mechanisms are involved in the pathophysiology of PE, such as the release of inflammatory cytokines, genetic predispositions, and immunological imbalance, which complicates the investigation of the underlying basal pathologic processes of this pregnancy disease [48,57]. Especially the dysfunctional remodeling of spiral arteries as well as the defective trophoblast invasion are important factors, that need to be further investigated to elucidate parts of the mechanism [21,48].

Since we have already shown a decrease of the histone modifications H3K4me3 and H3K9ac in PE in a prior publication [49], the aim of this study was to investigate a possible regulation of these histone modifications by proteins that seem to be involved in the pathology of PE. Therefore, we chose PPARγ which is known to play an important role in trophoblast differentiation and invasion [32]. Moreover, a connection of PPARγ with various histone modifications is already known in adipogenesis, and for example, the systemic lupus erythematosus [51,52,58], where a protective effect of PPARγ depending on histone acetylation has been shown [59]. Further, inhibition of histone deacetylase 3 was shown to lead to an increased expression of PPARy [60]. On the other hand, there are findings that point to a possible influence of PPARγ on histone modification, indicated by the experiments of Choi et al. [61] where an inhibitory effect of the histone demethylase KDM4D on adipogenesis could be shown. These effects could be rescued by adding exogenous PPARγ.

One of the most important known roles of PPARγ is the promotion of the transcription by heterodimerization with RxRα [33]. The heterodimer binds to the PPAR-response-element (PPRE) which is composed of two direct-repeat sequences, separated by one or two nucleotides in the promoter region of target genes [62].

In accordance with other publications [32,40], we detected a decreased expression of PPARγ and its binding heterodimer RxRα in the decidual tissue of PE-affected placentas. Additionally, we could detect a regulatory effect of PPARγ on the histone modifications H3K4me3 and H3K9ac. This indicates a pathophysiologic connection between PPARγ and the epigenetic modulation via H3K4me3 and H3K9ac in PE. Further, as PPARγ is implicated as a key regulator of trophoblast differentiation and invasion, which has been concretely shown in mice experiments [35] a decreased expression in the villous trophoblast of PE placentas could indicate defective placentation.

To confirm the association of PPARγ and the histone modifications in preeclampsia, which is known from other diseases, a correlation analysis was performed. The significant positive correlation of PPARγ expression with the histone modifications, as well as the co-expression in EVT, strengthened the idea of a possible regulation of the histone modifications through PPARγ.

To investigate this possible regulation, we performed cell culture experiments with HVT and primary isolated EVT cells. We examined the effect of the PPARγ agonist Ciglitazone and the PPARγ antagonist T0070907 on histone modification by immunocytochemical double immunofluorescence staining of H3K4me3 or H3K9ac and PPARγ.

Interestingly, the agonist Ciglitazone led to a reduced expression of PPARγ and histone modifications, while the antagonist T0070907 led to an increased expression of PPARγ and histone modifications. The interpretation of these results is diverse, but without doubt, a positive association of the expression of PPARγ and histone modifications can be established.

To adequately interpret the results, the in vivo effects of the agonists and antagonists must first be considered.

It has already been shown in animal studies that the inhibition of PPARγ by antagonists, for example, T0070907, leads to reduced fetal growth [36]. In contrast, activation of PPARy appears to protect against IUGR induced through hypoxia in advanced pregnancy [63]. In addition, as previously shown by Tache et al. [30], treatment with PPARγ antagonists can induce symptoms similar to those in preeclampsia. Moreover, treatment with a PPARγ agonist reduced RUPP-induced hypertension in an animal experiment using the reduced uterine perfusion pressure (RUPP) model [64], which suggests a protective role of PPARγ activity in endothelial cell function. Furthermore, the administration of PPARγ agonist supports villous cytotrophoblast differentiation. On the other hand, blocking PPARγ activation promotes proliferation and prevents differentiation of the villous trophoblast cells [20,65]. According to these findings, a decreased PPARγ-expression in the placental tissue would represent a reduced activation of PPARγ which leads to a defective trophoblast differentiation and invasion. The decreased mRNA level of PPARγ in PE placentas [32], as well as a reduction of circulating PPARγ activation which can be detected weeks to months before diagnosis [40], and our immunohistochemical staining results support this theory.

In contrast to this theory, other findings can be found in the literature, which support a hypothesis deduced from our cell culture results.

Corresponding to our findings, Levtyska et al. [20], who considered the activity of PPARγ in addition to the PPARγ expression, showed that stimulation of PPARγ activity with Rosiglitazone—a selective PPARγ agonist like Ciglitazone—led to a decreased PPARγ expression. Further, inhibition of PPARγ activity with the PPARγ antagonist T0070907 resulted in an increased PPARγ expression.

Barak et al. [66] showed that administration of Rosiglitazone led to changes of placental morphology and reduced the size of the placenta and the spongiotrophoblast layer. Further, EVT cell invasion is inhibited by PPARγ activation and improved by inhibition of PPARγ, shown in different cell culture models [34,67]. In addition, the treatment of mice with Rosiglitazone led to an altered microvasculature in the placenta and to a decreased expression of proangiogenic genes [68].

In summary, there seem to exist different effects of PPARγ stimulation on the different cell types of the placenta and their role during placentation.

PPARγ agonists and antagonists have moreover been studied in other models and biological systems than the placenta. In cancer cell models, inhibitory effects could be demonstrated on cancer cell growth by both, PPARγ agonists and PPARγ antagonists as well as apoptotic effects [69,70,71]. These findings show possible rectified effects of PPARγ agonists and antagonists in other cell models.

Our in vivo data underline this hypothesis as we could not find a clear difference of PPARγ-expression in the syncytiotrophoblast. In contrast, in EVT, the PPARγ-expression was decreased. Different studies have already shown an increase of LXRα, the PPARγ-target gene, in the tissue of PE placentas in accordance to an increased PPARγ-activity in PE [72,73]. Therefore, an increased PPARγ-activity might reduce PPARγ expression and vice versa, consistent to our in vitro data.

Prostaglandins which are known to be ligands of PPARγ, being increased in PE [74,75] might have a stimulating effect on the activity of PPARγ in EVTs, but this still needs to be verified in further studies and is merely part of a hypothesis (Figure 8).

The main limitation of this study is, that PPARγ activity has not been quantified in EVT cells, neither in vivo nor in vitro, and therefore should be part of further examination.

An additional limitation is the lower average week of gestation in the PE group, which needs to be considered in the interpretation of the results, although no statistical association was found.

Furthermore, no clear pathophysiologic statements of the development of PE could be concluded by this study. However, some interesting findings about epigenetic regulation during PE could be contributed.

## 4. Materials and Methods

### 4.1. Sample—Placental Tissue

A total number of 26 PE placentas and 25 placentas from healthy term pregnancies were analyzed. Mothers at delivery were between 17 and 44 years old (mean = 32.42 ± 5.929 years) and the week of gestation varied between the 25th and 40th (Appendix A).

Fetal sex was grouped into 25 females and 20 males, while for five placentas the gender declaration was missing. The equally distributed sex of the newborn was verified. Neither the age of the mother (*p* = 0.720), nor the sex (*p* = 0.692), or the weight of the newborn (*p* = 0.222) differed significantly between the groups of PE and control. Since the week of gestation differed significantly between the two groups (*p* < 0.001) regression analysis was performed to exclude the week of gestation as a potential confounder’ (Appendix A).

Placenta tissue was obtained directly after delivery in the Department of Obstetrics and Gynecology, LMU Munich between 2007 and 2019. Classification as preeclampsia was based on the guidelines of the German Society of Gynecology and Obstetrics. The tissue was collected by dissecting a piece of the placenta containing decidua and placental villi. The samples were immediately fixed in 4% neutral buffered formalin for one week, dehydrated, and embedded in paraffin for use in immunohistology and immunofluorescence.

The present study was approved by the local ethics committee of the Ludwig-Maximilians-University of Munich (reference number 18-700).

### 4.2. Immunohistochemical Peroxidase Staining

Immunohistochemical peroxidase staining was carried out according to the protocol used earlier [50]. After the blocking of endogenous peroxidase and heat pretreatment with a sodium-citrate-buffer (pH 6.0), an incubation with a blocking solution (ZytoChem Plus HRP Polymer System, mouse/rabbit; Zytomed Systems, Berlin, Germany) for five minutes, followed to avoid non-specific staining. Afterwards, the incubation with the primary antibody (anti-RxRα (Perseus Proteomics, Tokyo, Japan) or anti-PPARγ (Abcam, Cambridge, UK, antibody validation by manufacturer)) for 16 h at 4 °C in a humid chamber in the refrigerator (dilutions in Table 2) followed.

For visualization, the chromogenic 3,3′-diaminoenzidine (DAB; Dako, Glostrup, Denmark) was used. The staining reaction was stopped after a specific period for each primary antibody (RxRα 1 min, PPARγ 90 s) by adding distilled water. Counterstaining of the tissue was performed with Mayer’s acidic haematoxylin followed by dehydration in ascending alcohol series and a final Roticlear (Carl Roth GmBH, Karlsruhe, Germany) treatment, resulting in a brown color for positively stained cells and a blue color for negative cells.

Corresponding positive and negative controls can be found in Appendix A for antibody validation (Appendix A).

To analyze the slides the light microscope “Immunohistochemistry Type 307–148.001 512 686” by Leitz was used. Representative pictures were taken by IH-Camera 3CCD Colour Video Camera by Fissler. For image acquisition, the software “Discuss Version 4,602,017-#233 (Carl C. Hilgers Technical Office) was used. Time and space resolution data: 760 × 574 pixel.

### 4.3. Double Immunofluorescence Staining of Tissue Slides

The same primary antibodies and dilutions were used for the double immunofluorescence staining as described above (4.2.) except from PPARγ (LSBio, Seattle, DC, USA, dilution 1:500) and CK 7 (Novocastra, Wetzlar, Germany, dilution 1:30). The PPARy antibody was chosen for all immunofluorescence stainings despite the rabbit-host, as there was non-specific staining with several other antibodies which were tried.

The staining procedure differed depending on the hosts of the primary antibodies.

In the case of different hosts of the primary antibodies, the preparation of the tissue sections for immunofluorescence staining was performed similarly to immunohistochemical peroxidase staining. For detailed protocol see reference [50]. Secondary antibody was incubated for 30 min at RT (Cy3-labeled goat-anti-rabbit IgG—dilution 1:500 and Alexa Fluor 488-labeled goat-anti-mouse IgG—dilution 1:100) in Dako-dilution medium. After a final wash, the sections were covered with mounting medium containing 4′,6-Diamino-2-phenylindile (DAPI) as nucleic staining.

For the immunofluorescence double staining of PPARγ and the histone modifications, the hosts of the primary antibodies were the same, therefore the staining procedure had to be modified. After pretreatment of the samples similar to the immunohistochemical peroxidase staining, a first serum blocking was performed with 10%-goat serum for 45 min. Afterwards, an incubation with the first primary antibody (PPARγ) for one hour at room temperature followed. After a wash with PBS, the incubation with the first secondary antibody (Cy3-labeled goat-anti-rabbit IgG—dilution 1:500) for 30 min at room temperature followed. From this step on, it was necessary to work in the dark to avoid interference with the light-sensitive secondary antibodies. The next step included the second serum blocking, where normal 10%-rabbit serum was added for 45 min, to block the free binding sites of the anti-rabbit immunoglobulin and therefore to prevent unspecific staining. Afterwards, the sections were rinsed in PBS and the incubation with the second primary antibody (the respective histone-antibody) for one hour as well as incubation with the second secondary antibody (Alexa Fluor 488-labeled goat-anti-mouse IgG—dilution 1:100) for 30 min at room temperature, followed. After a final wash, the sections were covered with mounting medium containing 4′,6-Diamino-2-phenylindile (DAPI) as nucleic staining.

As a negative control for the staining, a specific IgG antibody was used in accordance with the utilized antibodies (Appendix A).

Pictures of the immunofluorescence staining were taken with Zeiss Axiophot fluorescence microscope (Zeiss, Oberkochen, Germany). For analysis the lens “40× CP-Achromat 40×/0.65, Infinity/0.17. Zeiss Part #44 09 50” was used. Dichroic filter cube sets from Omega for DAPI (UV Excitation, blue emission filter: Omega 365BP50; dichroic mirror: Omega 400DCLP, emission (barrier) filter: Omega 465DF60), FITC (Spectra: blue excitation, green emission, Omega cube set: XF100-2 with the following characteristics: excitation filter: 475/40, dichroic mirror: 505DRLP, emissions Filter: 535/45) and TRITC (Spectra: Green excitation, Red emissions; Omega dichroic filter cube: exciter: 525AF45, dichroic mirror: 560DRLP, emission (barrier) filter: 595AF60) were used.

The software AxioVision 4.8.1. was used to analyze the immune fluorescence staining. Image bit depth: 24 mm; time and space resolution data: 760 × 574 pixel. For the evaluation of the immunofluorescence staining intensity analysis of ZEN Software (Zeiss, Oberkochen, Germany) was used.

### 4.4. Staining Evaluation

The evaluation of the immunohistochemical peroxidase staining was performed by using the semi-quantitative Immunoreactive Score (IRS). In each case, the entire slide was evaluated, and the IRS was formed for syncytium and EVT individually. To calculate the IRS the staining intensity of the specific tissue (0 = no staining, 1 = weak staining, 2 = moderate staining, 3 = strong staining) was multiplied by the percentage of positively stained cells (0 = no staining, 1 = <10% of cells, 2 = 11–50% of cells, 3 = 51–80% of cells, 4 = >80% of cells stained) of the respective tissue. When forming the IRS, the quantification of the staining intensity is graded individually in each case according to the general staining intensity of the stain and is, therefore, a semi-quantitative evaluation. The evaluation of the staining was performed by two independent examiners.

### 4.5. Isolation of EVT Cells from Placenta Tissue

For the isolation of fresh EVT cells, placentas from healthy mothers were used directly after delivery. At first, the placenta was cut carefully, and the maternal part (decidua basalis) was separated from the fetal side (villous tissue). From here on, the maternal tissue was treated separately and washed several times with cold 0.9% NaCl (Carl Roth, Karlsruhe, Germany). After removing as much blood as possible, the last wash was performed with cold HBSS (Life Technologies, Carlsbad, CA, USA). The tissue was cut into small pieces and transferred in a glass bottle for digestion.

HBSS-HEPES-buffer was made from 25 mM HEPES (Sigma-Adrich, St. Louis, MO, USA) and HBSS (Life Technologies, Carlsbad, CA, USA) and the pH value was adjusted to 7.4. The DMEM (Sigma-Aldrich, St. Louis, MO, USA) was supplemented with 10% FBS superior (Sigma-Aldrich, St. Louis, MO, USA) and 1% antibiotic-antimycotic 100× (Life Technologies, Carlsbad, CA, USA). For the FACS staining buffer, 0.5% albumin (Carl Roth, Karlsruhe, Germany) and 2 mM EDTA (Sigma Aldrich, St. Louis, MO, USA) were dissolved in D-PBS (Life Technologies, Carlsbad, CA, USA).

Two digestion steps were performed to isolate EVTs. The first step was performed with 0.72 mg/mL trypsin (Sigma Aldrich, St. Louis, MO, USA) and 0.02 mg/mL DNAse1 (Roche, Basel, Switzerland) in HBSS-HEPES-buffer for 20 min at 37 °C. The second step was executed with 1 mg/mL Collagenase A (Roche, Basel, Switzerland) and 0.02 mg/mL DNAse1 in DMEM. After pulse centrifugation, the supernatant was layered on a Percoll (cytiva, Washington, DC, USA) gradient. For the preparation of the Percoll-tubes, 20%, 30%, 40%, 50%, 60%, and 70% Percoll-dilutions were set up with HBSS-HEPES-buffer. Afterwards the upper yellowish layer was removed and cells were washed twice with FACS-staining buffer. Lastly, the cells were counted with methylene blue (Stemcell Technologies, Vancouver, BC, Canada) in a Neubauer counting chamber, before being stored on ice.

### 4.6. FACS of Isolated EVT Cells from Placenta Tissue

To confirm the purity of primarily isolated EVT cells FACS analysis was performed. For intracellular trophoblast-specific CK 7 staining, permeabilization-buffer was made from 0.1% saponin (Carl Roth, Karlsruhe, Germany), 5% FBS superior (Sigma-Aldrich, St. Louis, MO, USA), and D-PBS (Life Technologies, Carlsbad, CA, USA). The Fc receptors were blocked in 10% human serum in D-PBS for 10 min at RT. For the surface staining, the cells were incubated in FACS-staining-buffer with CD45-FITC (BioLegend, San Diego, CA, USA) for 15 min at 4 °C. Then the cells were stained with fixable live/dead dye eFluor780 (Thermofisher, Waltham, MA, USA). Next, they were fixed with 1% PFA for 10 min at RT. For intracellular blocking, the cells were resuspended in permeabilization-buffer and 10% human sera was added for 10 min at 4 °C. Intracellular staining was executed with a CK 7-PE antibody (Abcam, Cambridge, UK) diluted in permeabilization-buffer. Before acquisition, the cells were washed in permeabilization-buffer and FACS-staining-buffer. Flow cytometry was performed on BD FACSCanto II and analyzed with FlowJo version 10.

### 4.7. Cell Culture of HVT-Cell Line and Primarily Isolated EVT Cells with PPARγ-Agonist and -Antagonist

To determine the influenceability of the histone modifications H3K4me3 and H3K9ac by PPARy, 50,000 HVT respectively primarily isolated EVT cells were seeded in 500 µL medium (RPMI-1640 + 10% FCS) per chamber of a chamber slide. After growing and adhesion to the slides, the cells were treated with the PPARγ-agonist Ciglitazone (20 mM, Tocris Bioscience, Bristol, UK) [69,70,76] and the PPARγ antagonist T0070907 (50 mM, Tocris Bioscience, Bristol, UK) or respective control vehicle. After an incubation period of 24 h, immunofluorescence staining was performed. The concentrations of the PPARγ-agonist Ciglitazone and the PPARγ antagonist T0070907, as well as the incubation period were chosen according to the literature published with these chemicals [69,70,71,76,77].

### 4.8. Double Immunofluorescence Staining of Chamber Slides

Fixation of the incubated cells in the chamber slides was carried out for 10 min in an ice-cold mixture of 100% methanol and ethanol in a relation of 1:1. After air-drying the slides, blocking, and staining of the primary antibody with combinations of H3K4me3, H3K9ac, and PPARγ (for dilutions see Table 1) was performed as described for the double immunofluorescence staining of tissue slides. Since no unspecific staining could be determined within the HVT-cells respectively isolated EVT, a PPARγ antibody with a mouse host was used (Abnova, Taipeh, Taiwan—dilution 1:100).

The slides were washed with PBS between all individual steps. Secondary antibody staining and covering was performed as for the double immunofluorescence staining of tissue slides. Corresponding negative controls can be found in the Appendix A. Representative pictures for analysis were taken with a confocal laser scan microscope (LSM 510 Meta, Zeiss). LSM 510 Meta 18 confocal laser scanning microscope consists of an Axiovert 200M equipped with Differential Interference Contrast (DIC) and with a range of excitation laser lines: Ar diode laser: 405 nm (30 mW), Ar: 458, 477, 488, 514 nm (30 mW), HeNe: 543 nm (1 mW), HeNe: 633 nm (5 mW). Used lens for analysis: 63× Plan-Apochromat NA 1.4, d in mm 0.19 from Zeiss.

### 4.9. Quantitative Analysis of Double Immunofluorescence Staining

For quantitative analysis of the staining intensity of the immunofluorescence staining of cultured trophoblasts, the ZEN Software (Zeiss, Oberkochen, Germany) was used. The mean fluorescence intensity of the investigated histone, receptor or protein was measured in three visual fields. The intensity of the concerned channel was evaluated per area (µm^2^) or set in relation to the DAPI channel to exclude the confounder of cell number in each visual field. The average of all visual fields was calculated and plotted in a diagram ±SEM.

### 4.10. Statistical Analysis

Statistical analysis was performed using SPSS (version 26 IBM company, Chicago, IL, USA) and GraphPad Prism (Version 6.0 GraphPad Software, La Jolla, CA, USA). Non-parametric tests were used for statistical analysis, due to not normally distributed data. Mann-Whitney-U-test was chosen for independent samples and Wilcoxon-signed-rank-test for paired samples. Results of these tests are given as mean value ± SD. For correlation analysis, Spearman–Rho correlation test was used. The correlation coefficient r indicates the strength of the correlation (r < 0.3 weak relation, r > 0.3 medium relation, r > 0.5 strong relation) [56]. For the double immunofluorescence staining of cultured trophoblasts, the *t*-test was used to examine differences between the two groups. The significance level for all tests was assumed at *p* < 0.05.

## 5. Conclusions

In summary, we showed that the expression of PPARγ regulates H3K4me3 and H3K9ac in HVT cells and primary isolated healthy EVT cells in a positive manner. Our in vivo results indicate that a reduced PPARγ expression correlates with the analyzed histone modifications and that an increased PPARγ activity might inhibit H3K4me3 and H3K9ac during PE. If considering our findings in relation to results from other studies, one can assume that a decreased expression of PPARγ is accompanied by increased activity. Thus, activation of PPARγ leads to a downregulation of the investigated histone modifications.

Whether H3K4me3 and H3K9ac act only as indicators for an abnormal trophoblast invasion through a lack of PPARγ-expression in PE, or whether they are responsible for defective placentation cannot be concluded from our results.

## Figures and Tables

**Figure 1 ijms-22-12469-f001:**
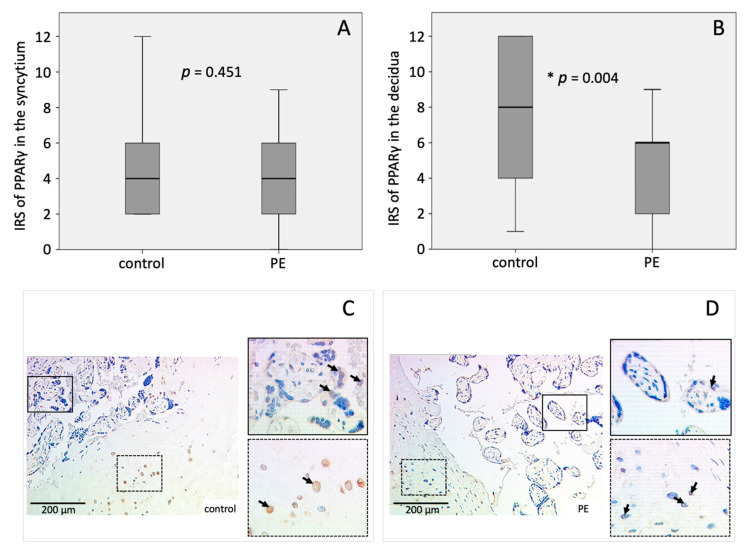
Immunohistochemical staining results of PPARγ. Boxplots of the IRS in the (**A**) syncytiotrophoblast and (**B**) the decidua as mean ± SD; *p*-values were calculated with Mann–Withney-U-Test; representative immunohistochemical images of PPARγ in controls (**C**) and PE placentas (**D**) were chosen, continuous line: syncytium, dotted line: decidua. IRS of control placenta: syncytiotrophoblast = 4, decidua = 4; IRS of PE placenta: syncytiotrophoblast = 4, decidua = 2. The respective *p*-value indicates, if the IRS of controls and PE differ significantly and was calculated using the Mann-Whitney-U-Test.

**Figure 2 ijms-22-12469-f002:**
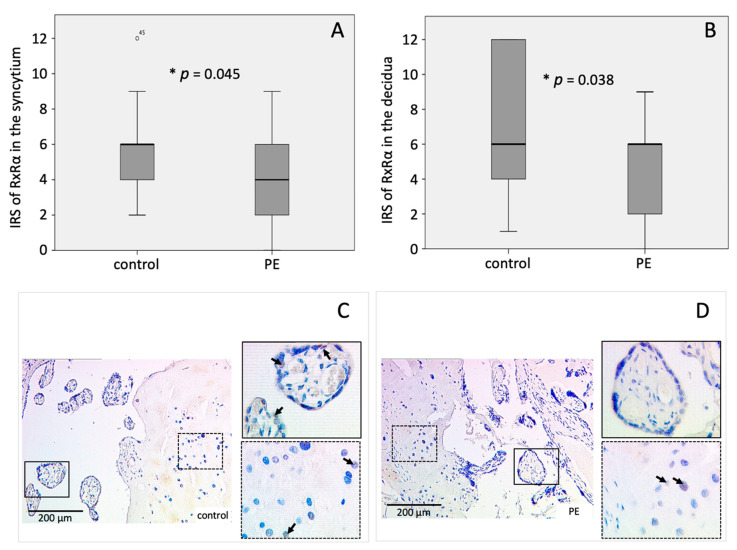
Immunohistochemical staining results of RxRα. Boxplots of the IRS in the (**A**) syncytiotrophoblast and (**B**) the decidua as mean ± SD; *p*-values were calculated with Mann–Withney-U-Test; representative immunohistochemical images of RxRα in controls (**C**) and PE placentas (**D**) were chosen, continuous line: syncytium, dotted line: decidua. IRS of control placenta: syncytiotrophoblast = 4, decidua = 3; IRS of PE placenta: syncytiotrophoblast = 1 decidua = 2. The respective *p*-value indicates, if the IRS of controls and PE differ significantly and was calculated using the Mann-Whitney-U-Test.

**Figure 3 ijms-22-12469-f003:**
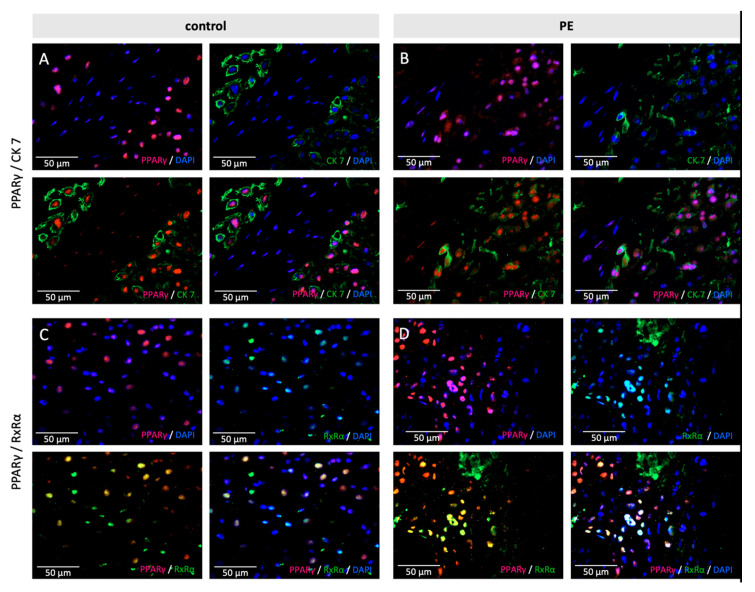
Examples of staining results of the immunofluorescence of PPARγ with CK7 (**A**,**B**) and RxRα (**C**,**D**), in control (**A**,**C**) and PE (**B**,**D**) placentas. Single immunofluorescence staining of PPARγ (pink). Single immunofluorescence staining CK7 or RxRα (green). Double immunofluorescence staining of PPARγ (**A**/**B**) and H3K4me3 (**C**/**D**) (red) and PPARγ (green). DAPI as nucleus staining (blue). Scale bar 100 µm.

**Figure 4 ijms-22-12469-f004:**
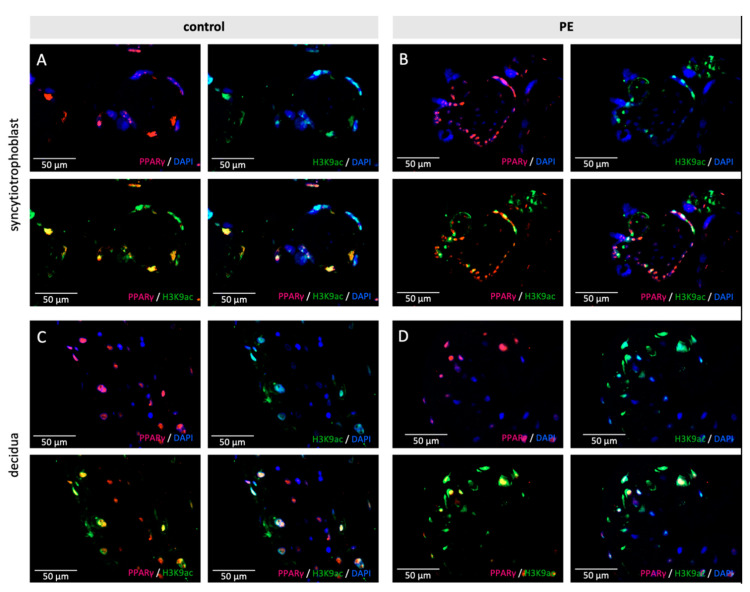
Examples of staining results of the immunofluorescence of PPARγ and H3K9ac (**A**,**B**)/H3K4me3 (**C**,**D**), in control and PE placentas. Single immunofluorescence staining of H3K9ac and H3K4me3 (pink). Single immunofluorescence staining of PPARγ (green). Double immunofluorescence staining of H3K9ac (**A**/**B**) and H3K4me3 (**C**/**D**) (red) and PPARγ (green). DAPI as nucleus staining (blue). Scale bar 100 µm.

**Figure 5 ijms-22-12469-f005:**
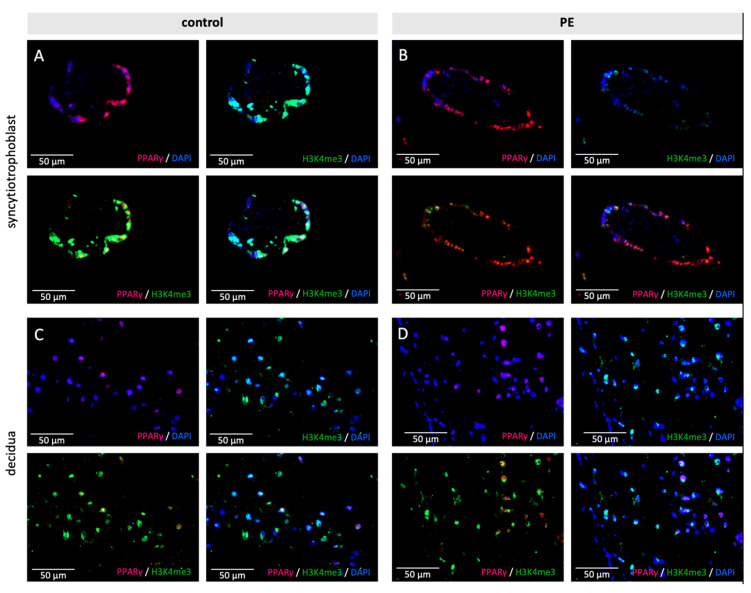
Examples of staining results of the immunofluorescence of PPARγ and H3K4me3 in control (**A**,**C**) and PE placentas (**B**,**D**) in the syncytiotrophoblast (**A**,**B**) and the decidua (**C**,**D**). Single immunofluorescence staining of PPARγ (pink) and H3K4me3 (green). Double immunofluorescence staining of H3K4me3 and PPARγ (yellow). DAPI as nucleus staining (blue). Scale bar 50 µm.

**Figure 6 ijms-22-12469-f006:**
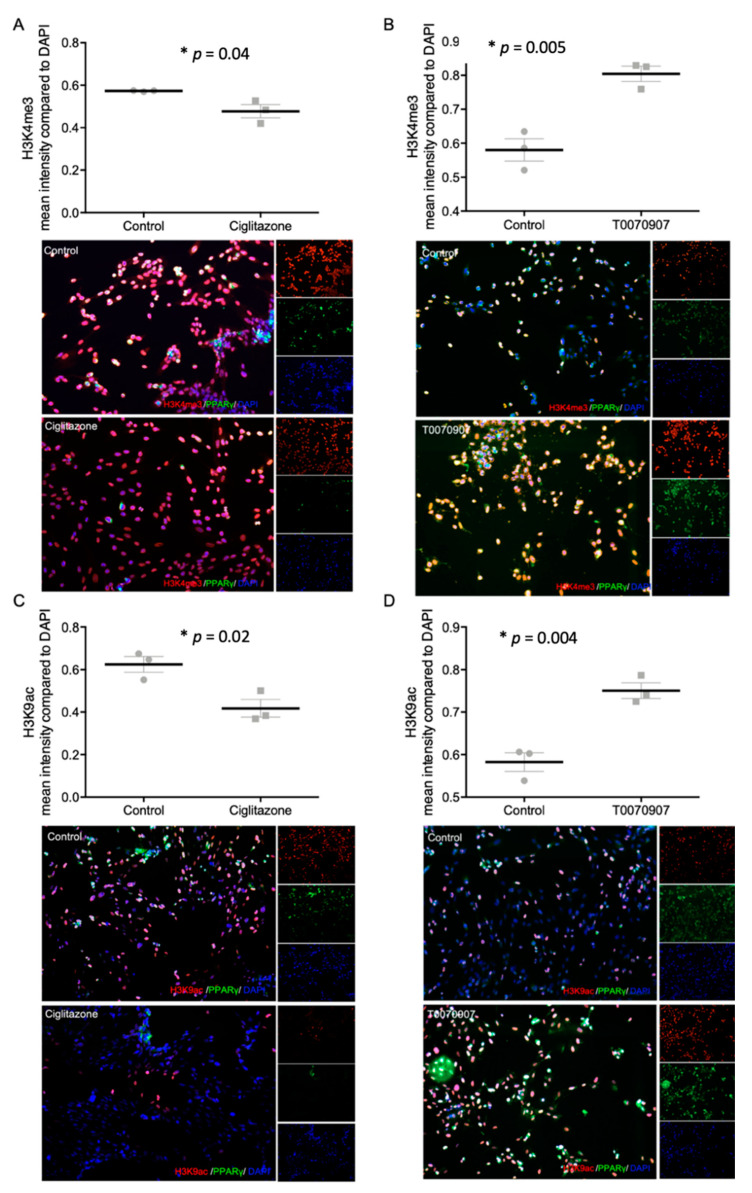
Staining results of histonemodifications H3K4me3 (**A**,**B**) and H3K9ac (**C**,**D**) and PPARγ after incubation with Ciglitazone (**A**,**C**) (20 mM) and T0070907 (**B**,**D**) (50 mM) mean fluorescence intensity in relation to DAPI, with representative pictures of EVTs, histone modifications shown in red, DAPI as nucleus staining in blue; Dot plots: mean fluorescence intensity ± SEM. The respective *p*-value indicates, if the intensity of control cells and cells incubated with Ciglitazone or T0070907 differ significantly and was calculated using the Mann-Whitney-U-Test. Images performed with 10× magnification, Scale bar: 50 µm.

**Figure 7 ijms-22-12469-f007:**
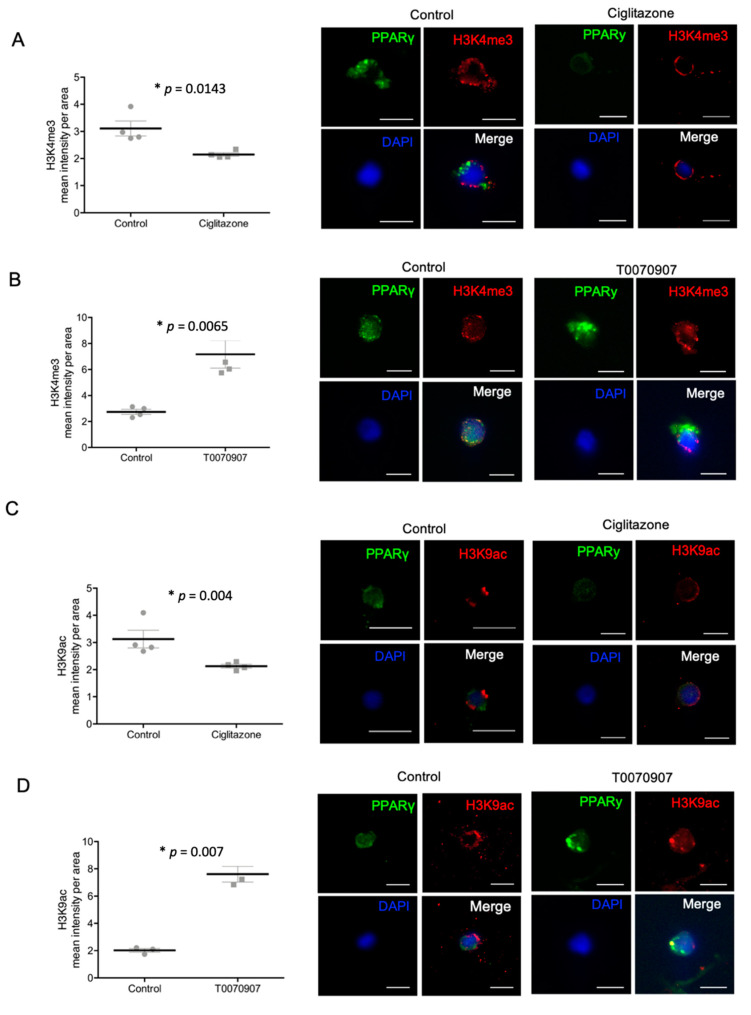
Staining results of histonemodifications H3K4me3 (**A**,**B**)/H3K9ac (**C**,**D**) and PPARγ after incubation with Ciglitazone (**A**,**C**) (20 mM) and T0070907 (**B**,**D**) (50 mM) mean fluorescence intensity per area, with representative pictures of isolated EVT cells, histone modifications shown in red, PPARγ in green, DAPI as nucleus staining in blue; Dot plots: mean fluorescence intensity ± SEM. The respective *p*-value indicates, if the intensity of control cells and cells incubated with Ciglitazone or T0070907 differ significantly and was calculated using the Mann-Whitney-U-Test. Images performed with 10× magnification, Scale bar: 10 µm.

**Figure 8 ijms-22-12469-f008:**
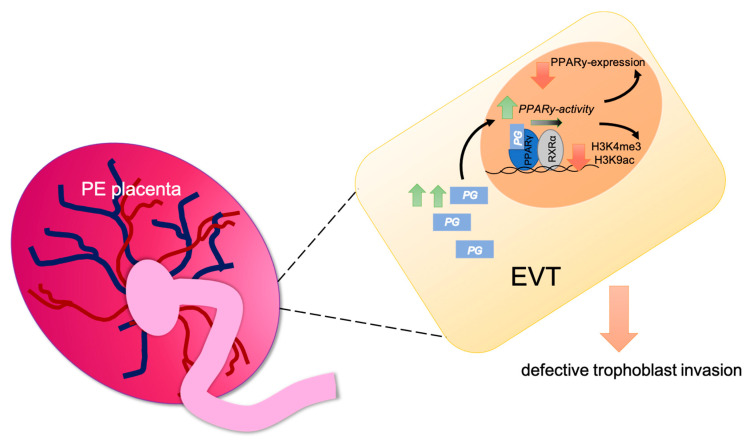
Graphical abstract of the hypothesis of PPARγ regulating H3K4me3 and H3K9ac in EVT (PG (prostaglandin)).

**Table 1 ijms-22-12469-t001:** Correlations of the histones with other proteins. The upper value is the correlation factor r, the second value is the *p*-value. * significant (*p* < 0.05), ** highly significant (*p* < 0.01) [56].

	H3K4me3		H3K9ac	
	Syncytium	Decidua	Syncytium	Decidua
RxRα syncytium	0.448 ** 0.002	0.513 ** 0.000	0.428 ** 0.003	0.577 ** 0.000
RxRα decidua	0.406 ** 0.007	0.327 * 0.032	0.389 ** 0.010	0.464 ** 0.002
PPARγ syncytium	0.216 0.131	0.249 0.084	0.284 * 0.046	0.379 ** 0.007
PPARγ decidua	0.357 * 0.012	0.227 0.121	0.347 * 0.014	0.399 ** 0.004

**Table 2 ijms-22-12469-t002:** Antibodies used in the immunohistochemistry (IH), immunofluorescence (IF) and immunocytochemistry (IC).

Antibody	Species Isotype	Manufacturer	Dilution	
H3K9ac	Rabbit IgG monoclonal (Clone: Y28)	Abcam	1:200	IH, IF, IC
H3K4me3	Rabbit IgG polyclonal	Abcam	1:100	IH, IF, IC
RxRα	Mouse IgG2a monoclonal (Clone: K8508)	Perseus Proteomics	1:200	IH, IF, IC
PPARγ	Rabbit IgG1 polyclonal	Abcam	1:100	IH
PPARγ	Mouse IgG1 monoclonal (Clone: 8D1H8F4)	Abnova	1:500	IC
PPARγ	Rabbit IgG polyclonal	LSBio	1:500	IF
CK 7	Mouse IgG1 monoclonal (Clone: NCL-L-CK7-OVTL)	Novocastra	1:30	IF
Cy2	Goat-Anti-Mouse IgG	Dianova	1:100	IF, IC
Cy3	Goat-Anti-Rabbit IgG	Dianova	1:500	IF, IC

## Data Availability

The datasets generated during the current study are available from the corresponding author on reasonable request.

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
