# Peer review of "Regulation of Epigenetic Modifications in the Placenta during Preeclampsia: PPARγ Influences H3K4me3 and H3K9ac in Extravillous Trophoblast Cells"

_ijms, 2021, doi:10.3390/ijms222212469_

Round 1
Reviewer 1 Report
The manuscript by Meister et al investigates the role of PPAR gamma in prececlampsia through regulating H3K4me3 and H3K9ac. The study builds on previously published work. The manuscript is well written and the data provided are preented clearly. The study uses PPAR gamma agonist and antagonist to show PPARgamma does in fact regulate H3K4me3 and H3K9ac, however the study is mostly observatory in nature, only showing changes in expression via histology. The study would be much more impactful if PPARgamma agonist and antagonists were used to show an alteration to also placental function - such as invasion and that these functional alterations are attributed to differences in H3K4me3 and H3K9ac.
Minor edit: line 64 - add such after comma so the text reads such as
Author Response
The manuscript by Meister et al investigates the role of PPAR gamma in prececlampsia through regulating H3K4me3 and H3K9ac. The study builds on previously published work. The manuscript is well written and the data provided are preented clearly.
Thank you very much for appreciating our study and manuscript.
The study uses PPAR gamma agonist and antagonist to show PPARgamma does in fact regulate H3K4me3 and H3K9ac, however the study is mostly observatory in nature, only showing changes in expression via histology. The study would be much more impactful if PPARgamma agonist and antagonists were used to show an alteration to also placental function - such as invasion and that these functional alterations are attributed to differences in H3K4me3 and H3K9ac.
Thank you very much for your feedback. It has already been shown in mice models that PPARγ is essential for the trophoblast invasion and other placental functions but yes, it would be very interesting to show that these alterations are due to changes in respective histone modifications, this has not been realized by our study. However, it would be interesting to build up on our results to perform e.g. animal experiments to show a causal relationship.
Minor edit: line 64 - add such after comma so the text reads such as
Thank you for this addition, we edited the manuscript according to your suggestions.

Reviewer 2 Report
The manuscript ijms-1455846 concerns the protein expression of PPARγ and RxRα in placenta tissue of preeclamptic patients. The topic is suitable to the journal. The methodology of the study involves a series of immunostainings. However, several parts of the manuscript require revision in order to add clarity or to make the manuscript better focused. I have listed some concerns/suggestions below:
- I suggest to transfer the conclusions section after discussion.
- Why r>0.5 was defined as a strong correlation? Usually correlation coefficients whose magnitude are greater than 0.7 indicate variables which can be considered strongly correlated.
- Lines 151-153 – please add literature to confirm the sentence.
- Mechanistic studies have not been carried out. More research is needed to determine the role of PPARγ in H3K4me3 and H3K9ac regulation.
- Based on the study, in my opinion it is incorrect to conclude that PPARγ regulates H3K4me3 and H3K9ac. It can be assumed that PPARγ is involved in the regulation of H3K4me3 and H3K9ac. Title of the manuscript should be improved.
- How the concentrations of agonists and antagonists were selected? Did the Authors confirm its action at doses mentioned in the manuscript?
- Line 332- ‘for eight placentas the gender declaration was missing’ – According to Suppl. Tab. 1 gender declaration was missing in seven patients – please correct
- Did the groups differ in case of mother’s age?
Author Response
The manuscript ijms-1455846 concerns the protein expression of PPARγ and RxRα in placenta tissue of preeclamptic patients. The topic is suitable to the journal. The methodology of the study involves a series of immunostainings.
However, several parts of the manuscript require revision in order to add clarity or to make the manuscript better focused.
Thank you very much, we appreciate reviewer´s suggestions and tried to refer to your comments and concerns to improve our manuscript.
I have listed some concerns/suggestions below:
- I suggest to transfer the conclusions section after discussion.
We transferred the conclusion section after discussion. If that is in agreement with the style of the journal we would keep this.
Why r>0.5 was defined as a strong correlation? Usually, correlation coefficients whose magnitude are greater than 0.7 indicate variables which can be considered strongly correlated.
Following J. Cohen, a pioneer of statistics, this assessment of the correlation coefficient was chosen. The reference was added in our manuscript. We hope that our argumentation is acceptable for the reviewer, if not please let us know.
Lines 151-153 – please add literature to confirm the sentence.
We added literature to confirm the sentence.
Mechanistic studies have not been carried out. More research is needed to determine the role of PPARγ in H3K4me3 and H3K9ac regulation.
We totally agree with the reviewer that there is more research needed to determine the role of PPARγ in the regulation of H3K4me3 and H3K9ac. Our study is providing important finding about the role of PPARγ in regulating H3K4me3 and H3K9ac but only small parts of the complete story
- Based on the study, in my opinion it is incorrect to conclude that
PPARγ regulates H3K4me3 and H3K9ac. It can be assumed that PPARγ is
involved in the regulation of H3K4me3 and H3K9ac. Title of the manuscript should be improved.
Thank you very much for this important annotation. We tried to improve and to adapt the title more suitable to our results and tried to formulate more carefully. We did not want tp irritate the readers and agree with you that PPARγ is merely involved in the regulation of H3K4me3 and H3K9ac through a regulation cascade, which has not been cleared in detail by our results.
How the concentrations of agonists and antagonists were selected? Did the Authors confirm its action at doses mentioned in the manuscript?
We chose Ciglitazone and T0070907 as PPARγ-agonist and -antagonist because they are specifically binding to PPARγ and are very potent Please refer to respective datasheets: https://documents.tocris.com/pdfs/tocris_coa/1307_1_coa.pdf?1636730116&_ga=2.88931611.145420636.1636729271-232598621.1636729271; https://documents.tocris.com/pdfs/tocris_coa/2301_1_coa.pdf?1636729268&_ga=2.68073745.145420636.1636729271-232598621.1636729271)
We tried to explain why we selected respective PPARγ agonist and antagonist concentrations and incubation times and for our study in the methods. We hope we could satisfy the reviewer with our explanation, if not we will try to improve it concerning reviewers wishes.
Line 332- ‘for eight placentas the gender declaration was missing’ – According to Suppl. Tab. 1 gender declaration was missing in seven patients – please correct
Thank you for drawing attention to this failure. We corrected the number of missing gender declarations in the manuscript.
- Did the groups differ in case of mother’s age?
The age of the mothers differed not significantly between the groups. We added information in our manuscript in line 341 and 342

Reviewer 3 Report
In this manuscript Meister and colleagues found a reduced expression of PPARγ and RxRα and a reduced co-expression with H3K4me3 and H3K9ac in the extravillous trophoblast. Furthermore, they found that the stimulation of PPARγ-activity leads to a reduction of H3K4me3 and 37 H3K9ac in trophoblast cells but also a decrease of the nuclear PPARγ expression.
This is a well written and interesting study. However some points need to be clarified:
- a summary table with clinical parameters of patients (proteinuria, blood pressure, placental and neonatal weight) must be provided. Moreover, three PE samples with unknown gestational ages cannot be used.
- Line 347. the reference "Hutter et al., 2015b" should be modified accorting to the journal style.
- Materials and methods: How did the authors quantify the Immunohistochemical staining (Figures 1 and 2)? Rapresentative images of 0 = no staining, 1 = weak staining, 2 = moderate staining and 3 = strong staining should been shown. Have it been performed by at least two operators?
- Figure 3C; PPARγ/RxRα staining ... both antibodies are made in mouse and have been used to co-localize PPARγ and RxRα in the nuclei. However, they used two anti-mouse secondary antibodyies that can crossreact with both PPARγ and RxRα primary antibodies made in mouse. This is not the optimal setting to colocalize two proteins in the same compartment (nuclei) because the secondary antibodies can stain both PPARγ and RxRα primary antibody. Why the authors did not use an antibody from a different species?
Author Response
In this manuscript Meister and colleagues found a reduced expression of
PPARγ and RxRα and a reduced co-expression with H3K4me3 and H3K9ac in
the extravillous trophoblast. Furthermore, they found that the stimulation of PPARγ-activity leads to a reduction of H3K4me3 and 37 H3K9ac in trophoblast cells but also a decrease of the nuclear PPARγ expression.
This is a well written and interesting study.
Thank you very much for appreciating our study results and manuscript. We tried to follow your comments and your suggestions.
However some points need
to be clarified:
1. a summary table with clinical parameters of patients (proteinuria,
blood pressure, placental and neonatal weight) must be provided.
Moreover, three PE samples with unknown gestational ages cannot be used.
We are very sorry for not being able to provide all the missing data about blood pressure, blood values, placental and neonatal weight, since these data have not been registered in detail during recruitment or cannot be retraced since data have been deleted because of data privacy protection. We provided all data which was available.
The diagnosis of preeclampsia was made according to the guidelines of the German association of gynecology and obstetrics (DGGG). Accordingly, neither the pure level of proteinuria nor the week of gestation was taken as decisive criteria for the diagnosis. Therefore, we would not like to exclude all cases in which the data of the week of gestation or proteinuria were missing from our analysis. We hope the reviewer finds our way of argumentation comprehensible and hope that he accepts that and that we could take reviewers doubts by our argumentation. If not please let us know.
Line 347. the reference "Hutter et al., 2015b" should be modified
according to the journal style.
Thank you very much, we modified the reference according to the journal style.
Materials and methods: How did the authors quantify the Immunohistochemical staining (Figures 1 and 2)? Representative images of 0 = no staining, 1 = weak staining, 2 = moderate staining and 3 = strong staining should been shown. Have it been performed by at least two operators?
We provided representative images in our supplement in Supp. Table 2. Yes, the IRS has been performed by two independent operators.
Figure 3C; PPARγ/RxRα staining ... both antibodies are made in mouse
and have been used to co-localize PPARγ and RxRα in the nuclei. However, they used two anti-mouse secondary antibodyies that can crossreact with both PPARγ and RxRα primary antibodies made in mouse.
This is not the optimal setting to colocalize two proteins in the same compartment (nuclei) because the secondary antibodies can stain both PPARγ and RxRα primary antibody. Why the authors did not use an antibody from a different species?
Thank you very much for this annotation. We clarified in our manuscript which antibody was used for which staining and how we avoided an unspecific staining despite the same host. We tried several antibodies against PPARγ from different hosts, but the used one for IF double staining was the most specific one. We clarified unspecific cross reaction before and performed different steps of blocking. Therefor we hope that reviewer finds our argumentation for choosing this antibody acceptable.
